# Cannabis: Zone Aspects of Raw Plant Components in Sport—A Narrative Review

**DOI:** 10.3390/nu17050861

**Published:** 2025-02-28

**Authors:** Corina Flangea, Daliborca Vlad, Roxana Popescu, Victor Dumitrascu, Andreea Luciana Rata, Maria Erika Tryfon, Bogdan Balasoiu, Cristian Sebastian Vlad

**Affiliations:** 1Department of Biochemistry and Pharmacology, Faculty of Medicine, “Victor Babeș” University of Medicine and Pharmacy, 2nd Eftimie Murgu Square, 300041 Timisoara, Romania; flangea.corina@umft.ro (C.F.); dumitrascu.victor@umft.ro (V.D.); vlad.cristian@umft.ro (C.S.V.); 2Toxicology and Molecular Biology Department, “Pius Brinzeu” County Emergency Hospital, Liviu Rebreanu Boulevard 156, 300723 Timisoara, Romania; popescu.roxana@umft.ro; 3Department of Cell and Molecular Biology, Faculty of Medicine, “Victor Babeș” University of Medicine and Pharmacy, 2nd Eftimie Murgu Square, 300041 Timisoara, Romania; 4Surgical Emergencies Department, “Victor Babeș” University of Medicine and Pharmacy, 2nd Eftimie Murgu Square, 300041 Timisoara, Romania; andreea.rata@umft.ro; 5Faculty of Medicine, “Victor Babeș” University of Medicine and Pharmacy, 2nd Eftimie Murgu Square, 300041 Timisoara, Romania; maria.tryfon@t-online.de (M.E.T.); bogdan.balasoiu@student.umft.ro (B.B.)

**Keywords:** cannabinoids, tetrahydrocannabinol, cannabidiol, cannabigerol, sport doping, health risk, therapeutic effects

## Abstract

**Objectives/Background:** The Cannabis genus contain a mixture of cannabinoids and other minor components which have been studied so far. In this narrative review, we highlight the main aspects of the polarized discussion between abuse and toxicity versus the benefits of the compounds found in the *Cannabis sativa* plant. **Methods**: We investigated databases such as PubMed, Google Scholar, Web of Science and World Anti-doping Agency (WADA) documents for scientific publications that can elucidate the heated discussion related to the negative aspects of addiction, organ damage and improved sports performance and the medical benefits, particularly in athletes, of some compounds that are promising as nutrients. **Results**: Scientific arguments bring forward the harmful effects of cannabinoids, ethical and legislative aspects of their usage as doping substances in sports. We present the synthesis and metabolism of the main cannabis compounds along with identification methods for routine anti-doping tests. Numerous other studies attest to the beneficial effects, which could bring a therapeutic advantage to athletes in case of injuries. These benefits recommend *Cannabis sativa* compounds as nutrients, as well as potential pharmacological agents. **Conclusions and Future Perspectives**: From the perspective of both athletes and illegal use investigators in sport, there are many interpretations, presented and discussed in this review. Despite many recent studies on cannabis species, there is very little research on the beneficial effects in active athletes, especially on large groups compared to placebo. These studies may complete the current vision of this topic and clarify the hypotheses launched as discussions in this review.

## 1. Introduction

The use of banned substances to enhance athletic performance is a hot topic. With the legalization of the recreational use of cannabinoids in some countries, discussions have arisen regarding their use among athletes, with opinions both for and against their use. Currently, cannabinoids are on the World Anti-doping Agency (WADA) list of substances prohibited in sport only during competitions [1]. Among these cannabinoids, the exception is cannabidiol (CBD), which was removed from the list of prohibited substances initially in 2018 [2]. Moreover, starting 2013, WADA modified the minimum permitted concentrations of Δ^9^-tetrahydrocannabinol (THC) in urine, increasing them 10-fold, from 15 ng/mL to 150 ng/mL [2]. Going further, Major Baseball League removed marijuana from its list of prohibited substances in sports, adopting a similar attitude toward its use as it has toward alcohol consumption [3]. Negotiations continue, also considering the beneficial effects of cannabinoids in various pathological conditions, with some authors considering that cannabinoids are not doping substances [4].

Cannabis use in sports is a common but not always reported practice among high-performance athletes, with an estimated one in four athletes using it [5]. Several reviews have attempted to provide an overview of the frequency of use in sports. Among the results that stand out are the high percentages of athletes who reported using marijuana in the last 30 days: American college athletes (11.2%), French high school athletes (19%), with the highest percentage (32.7%) being French athletes participating in regional, national, and international competitions [6]. The National Collegiate Athletic Association reports that 24% of athletes are marijuana users [7]. Regarding the proportion of consumers in each sport, the percentages of consuming athletes are somewhat similar, regardless of the type of sport practiced: 24.8% in runners, 22.8% in cyclists and 24.2% in triathletes [8]. Although these studies and reports attempt to describe the current situation, the true extent of consumption of marijuana or cannabinoids extracted from *Cannabis sativa* will be known with greater accuracy only when prohibition is lifted. In these conditions, athletes will no longer fear the legal repercussions that may fall upon them if they declare this consumption.

On the other hand, the consumption of cannabinoids, especially CBD, has shown promise in chronic musculoskeletal diseases occurring post-trauma or due to overtraining, as happen in sports such as tendon injuries, chronic back pain, joint injuries, which do not respond satisfactorily to nonsteroidal anti-inflammatory drugs [9]. Participants in a study where they ran 3.88 miles reported a lower incidence of pain after acute cannabis use [10].

The introduction of CBD dietary supplements is currently being considered [11,12]. Furthermore, marijuana products are considered nutriceuticals for their analgesic effect, emphasizing phytomedicine as a therapeutic option in migraine [13]. However, its nutritional properties have been recently reviewed [14,15,16]. Due to these high nutritional values, some authors even go so far as to consider it for possible use in vegetarian diets [15]. Another utility in athletes who have a heavy competitive season is the demonstrated effectiveness of CBD in cases of stress-induced illnesses [17]. But these aspects cannot be fully extrapolated to high-performance athletes.

In this narrative review, we provide a discussion of the scientific arguments on the harmful (dark zone) vs. beneficial (light zone) effects of phytocannabinoids along with other compounds found naturally in the *Cannabis sativa* plant. Besides its well-known appearance, presented in Figure 1, a discussion is also necessary regarding the main categories of substances synthesized in this famous plant. As we might expect, the line between the two zones is very fragile, and depending on the context, the balance can tip to one side or the other.

## 2. Methods

In this narrative review, we used the databases PubMed, Google Scholar and Web of Science as well as WADA documents, and selected the most relevant papers for the purpose of this review. We used as keywords “cannabinoids”, “sport”, “illegal use”, “tetrahydrocannabinol”, “cannabidiol”, “cannabigerol”, “myrcene”, “limonene”, “pynene”, “sesquiterpenoids”, “friedelin”, “sitosterol”, “cannflavins”, “pharmacokinetics” and combinations of them. The toxic, moral and illegal effects (dark zone) of cannabinoid consumption by athletes were discussed in parallel with the beneficial effects (light zone) and the possibility of using these compounds as nutrients. Between the two opposite zones, sensitive and questionable aspects of this consumption (grey zone) were also touched upon.

## 3. Main Components Found in *Cannabis sativa* Plant

*Cannabis sativa* has been of interest since ancient times due to the numerous pharmacologically active compounds found in it. This plant contains a complex mixture belonging to several classes of substances such as cannabinoids, flavonoids, sterols, monoterpenoids, triterpenoids, and sesquiterpenoids. Depending on the class of compounds, there are several metabolic pathways of production and interconversion: (i) polyketide (PK) pathway, (ii) methyl-erythritol phosphate (MEP) pathway, (iii) mevalonate (MV) pathway, (iv) phenyl-propanoid (PP) pathway [18].

### 3.1. Polyketide (PK) Pathway

Polyketide (PK) pathway is the synthesis pathway of cannabinoids that starts from activated hexanoic acid in the form of hexanoyl-CoA [19,20,21,22,23,24,25]. The first step of this pathway begins with a condensation between hexanoyl-CoA and malonyl-CoA residues to form a 12-carbon tetraketide intermediate, under the action of the enzyme tetraketide synthase (TKS). This structure undergoes either cyclization by olivetolic acid cyclase (OAC) to form olivetolic acid, or cyclization and decarboxylation by TKS to form olivetol. In the second step, olivetolic acid condenses with geranyl-diphosphate, a common precursor of the MEP and MV pathways, the reaction being catalyzed by cannabigerolic acid synthase (CNGAS). A large molecule, cannabigerolic acid, is formed, part of which undergoes spontaneous decarboxylation to cannabigerol. From cannabigerolic acid, two categories of compounds are derived: Δ^9^-tetrahydrocannabinolic acid via tetrahydrocannabinol synthase (THCS) and cannabidiolic acid via cannabidiolic acid synthase (CNDAS). Under the action of heat, light, and UV radiation, both compounds are decarboxylated, resulting in Δ^9^-tetrahydrocannabinol (THC) and cannabidiol (CBD) (Figure 2).

### 3.2. Methyl-Erythritol Phosphate (MEP) Pathway

Methyl-erythritol phosphate (MEP) pathway has as its final products monoterpenes, of which the best known are β-myrcene, α-pynene and limonene, but also a multitude of minor compounds such as camphene, α-terpineol, terpinene, and β-pinene [26,27,28,29,30,31,32]. Briefly, the process begins by coupling glyceraldehyde-3-phosphate with a molecule of pyruvic acid, followed by decarboxylation and formation of deoxy-xylulose-5-phosphate. In a subsequent step, hydroxy-methyl-butenyl-diphosphate (HMBPP) is formed, passing through 2-methyl-erythritol-4-phosphate and methyl-erythritol-phosphate (MEP). From HMBPP, dimethyl-allyl-diphosphate (DADP) will be produced, which will isomerize to isopentenyl-diphosphate (IPDP), followed by the formation of geranyl-diphosphate. Geranyl-diphosphate can be used in the synthesis of cannabinoids via the PK pathway or can further be converted to monoterpenes [18,27,29,32] (Figure 3).

### 3.3. Mevalonate (MV) Pathway

Mevalonate (MV) pathway begins with a phase that has DADP and its isomer, IPDP, as intermediate products, but their production follows different steps compared to the MEP pathway. Initially, mevalonate is formed from the condensation of three molecules of acetyl-CoA. Once IPDP has been produced, the dynamics of IPDP isomerization to DADP will evolve towards the formation of farnesyl-diphosphate, a substance from which differentiation begins into three different pathways: sesquiterpenoids, triterpenoids and sterols [27,33,34].

Sesquiterpenoids identified in *Cannabis sativa* are formed directly from farnesyl-diphosphate. Several authors have identified numerous sesquiterpenes, the most abundant of which were the following: β-caryophyllene, α-selinene, β-farnesane, curcumene and α-caryophyllene [30,35,36]. Triterpenes and sterols require an additional transformation of farnesyl-diphosphate to squalene, from which the two categories of substances are then derived. Among the triterpenoids, friedelin and epifriedelanol have been found as major compounds [37,38,39,40] while the sterols found are mainly β-sitosterol, campesterol, and stigmasterol [38,39,40,41] (Figure 4).

### 3.4. Phenyl-Propanoid (PP) Pathway

Phenyl-propanoid (PP) pathway is a flavonoid synthesis pathway that starts from phenylalanine. A wide range of flavonoids specific or non-specific to *Cannabis sativa* can be synthesized in this way. Cannflavins, the specific flavonoids, are produced starting from phenylalanine. This is transformed into p-cinnamic acid, the reaction being catalyzed by phenylalanine-ammonia-lyase. An OH group in the para position is introduced by the enzyme cinnamate-4-hydroxylase resulting in p-coumaric acid, which is then activated to p-coumaroyl-CoA by 4-coumarate-CoA-ligase. Three molecules of malonyl-CoA will condense on this compound forming naringeninchalcone which will isomerize to naringerin under the action of chalone-isomerase. In a subsequent step, flavone synthase will act with the formation of apigenin; apigeninis are transformed into luteolin with contribution of flavonoid-3-hydrolase, which will then form chrysoeriol, a methyl ether produced through the action of O-methyl-transferase, a specific enzyme of *Cannabis sativa*. In a final step, chrysoeriol is converted into cannflavins, also with the support of a specific *Cannabis sativa* enzyme, prenyl-transferase [42,43,44] (Figure 5).

Among the nonspecific flavonoids, some authors have identified numerous compounds, the most important of which seem to be kaempferol, quercetin, myricetin, vitexin and isovitexin [44,45,46]. Their synthesis is common with that of cannflavins until the formation of naringenin, which will be transformed under the action of flavone-3-hydroxylase whence the production pathway of dihydrokaempferol, dihydroquercetin, or dihydromyricetin will follow, resulting in kaempferol, quercetin and myricetin, respectively [45].

## 4. Dark Zone: Abuse and Prohibition in Sport

Cannabinoids and their synthetic derivatives are among the substances prohibited in sports because they meet three important and mandatory conditions: (i) direct contribution to increasing sports performance; (ii) harmful effects on the body; (iii) violation of the spirit and philosophy of sports [5].

The first condition refers to the improvement of sports performance in terms of both strength and endurance and increased pain tolerance. Among the effects that contribute to this aspect are vasodilation and bronchodilation [47,48,49]. Also, through their anxiolytic and antidepressant effects, they allow athletes to function much better under stressful conditions during and before competition [49,50,51]. These advantages are especially welcome in sports where a greater capacity for concentration is required. These effects also contribute to better sleep [52]. Added to this is the analgesic effect that allows athletes to overcome minor injuries and those caused by fatigue during exercise [47].

The second condition refers to the toxic effects produced by cannabinoids. One such effect is addiction: even though no specific patterns of physical dependence have been identified, alterations in emotional and cognitive function with extended negative symptoms have been described in chronic users, even after a period of 28 days after cessation of use [53,54]. Psychological dependence is characterized by excessive impulsivity to consume the drug and is accompanied by hyper-reactivity of the mesocorticolimbic dopaminergic reward system [55]. In the brain, cannabinoids affect short-term memory with a reduction in the ability to learn and retain new information. Motor coordination is also affected. In the long term, cannabis use increases the incidence of developing paranoia, schizophrenia, or other types of psychosis [56,57]. At the cardiovascular level, cases of acute myocardial infarction in the absence of vascular risk factors have been described after marijuana use [58]. Stimulation of the sympathetic autonomic nervous system and inhibition of the parasympathetic autonomic nervous system contribute to this effect. Other events reported were arrhythmias: paroxysmal tachycardia, atrial flutter, atrial fibrillation, and ventricular arrhythmias [59]. Cases of cannabis-induced cardiomyopathy have also been described [60]. At the pulmonary level, an increase in the incidence of chronic obstructive pulmonary disease, chronic bronchitis, pulmonary emphysema and idiopathic pulmonary fibrosis exacerbated by concomitant smoking of cigarettes with tobacco has been observed. An important contribution to these changes is made by combustion gases and carbon monoxide resulting from marijuana smoking [60,61]. These effects have been observed in chronic users despite the bronchodilator effect of acute consumption [61].

The third condition is a condition of life ethics in general, and sports ethics in particular [47,52]. Athletes usually represent successful models, worthy of following in life. This educational character contradicts drug abuse and cheating. On the other hand, in major competitions where certain amounts of money obtained as prizes by the winners are involved, falsifying the result in this way contradicts the law. In other words, it can be said that these types of prizes are stolen from an athlete, who, by his or her own means, obtains certain results, by another athlete who “wins” by means of the substances of abuse.

### 4.1. A Brief Discussion of Current Worldwide Legislative Implications

Although the use of cannabinoids recreationally or for various medical reasons has a lengthy history, most often “through the back door”, it is currently under dynamic discussion in terms of changing legislation in most countries.

From a medical point of view, starting in 2019, the European Medicines Agency approved CBD (cannabis extract) as the first product for the treatment of epilepsy in children who do not respond adequately to conventional treatment [62]. In many European countries (but not in Bulgaria, Cyprus, Greece, Hungary, Latvia, Romania, or Slovakia) the use of THC and CBD as cannabis extracts under the name Nabiximols, a sublingual spray in the treatment of multiple sclerosis to relieve muscle spasm, was legalized before this directive. Some European countries accept the use of official prescriptions (Germany, Italy, The Netherlands, Croatia, Luxembourg, Poland, Sweden, Czech Republic) or magistral prescriptions (Germany, Italy, The Netherlands, Czech Republic) for medical purposes [63]. But this is just the beginning, where things start to change.

From the point of view of recreational use, the relaxation of legislation and the permission of consumption have attracted many alarm bells, especially by studies carried out in the United States. These studies mainly target adolescents, young adults, but also children. The most dangerous effects have been those produced by accidental overdose, but also by the ingestion of these preparations by children, increasing the number of presentations in the emergency department of pediatric hospitals by 57% [64]. Other worrying aspects were represented by the appearance of psychiatric symptoms, but also by some crimes or car accidents committed by people who had consumed cannabis [65]. In April 2024, a European Parliamentary directive attempted to set limits on recreational use within the international drug control system. These directives ranged from legalizing use, as in Germany, to banning use, as in Bulgaria [66]. An overview of the situation in the world is briefly summarized in Table 1.

It can be seen that, with the legal use of cannabinoids for medical purposes, there has also been an increase in recreational use. In this context, many countries have been somewhat “constrained” to change their legislation and relax it regarding the use of cannabis. With the lifting of legislative barriers, there has been an increase in use among children and adolescents; however, on the other hand, the black market in cannabinoids is discouraged.

### 4.2. Important Pharmacokinetic Processes in Cannabinoid Detection

The identification and quantification of components in *Cannabis sativa* is influenced by their pharmacokinetics in terms of the interval and detection period from administration, the route of administration, the biological matrix subjected to analysis and the resulting metabolites.

Bioavailability and the time to reach maximum plasma concentration depend on the route of administration and the first hepatic passage (for the oral route), athletes being usually healthy individuals in whom there are usually no interferences related to decreased liver function and tissue hypoperfusion. After oral administration, absorption is good due to the high liposolubility of most compounds. However, THC and CBD have a low bioavailability, estimated at 6% [78,79], due to the first-pass effect in the liver. Also, the latency and time to reach maximum plasma concentration are long, 2 h for THC [80] and 1–6 h for CBD [78]. Transdermal administration is a preferred route for local applications and less frequently for systemic effects. However, studies have shown that peak plasma concentrations of CBD and THC occur after 8 h [81], with bioavailability that can be increased in the presence of enhancers [82]. Although this route of administration is less commonly used recreationally, it should be considered when there is a recommended pathology or intentional abuse. Thus, the most intense and rapid effect is obtained after inhalation administration, which is a common approach for recreational use or when a quick effect is desired.

The steady state volume of THC is around 2.5–10 L/kg [78]. Cannabinoids, being liposoluble compounds, have a high capacity to accumulate in fat-rich tissues, following a biphasic process. Initially, they are distributed in highly vascularized tissues such as liver, kidney, heart, brain, followed by a second redistribution to less vascularized tissues with a rich lipid content such as adipose tissue. With prolonged use, a balance is created between elimination and administration, by gradual release from stores, sometimes for several weeks after stopping consumption [78,79,80]. From a toxicological point of view, accumulation in nails and hair is also important, where consumption could be documented retrospectively [83].

From a detection point of view, THC metabolism is of major practical importance. Both THC and CBD undergo hepatic metabolism, 80–90% being eliminated as hydroxylated and carboxylated metabolites in urine and feces. The first compound, 11-hydroxy-THC, is an active metabolite that will subsequently be transformed into 11-nor-9-carboxy-THC, passing through an intermediate, 11-oxo-THC. These transformations are catalyzed by CYP2C9, CYP2C19 and CYP3A4 [84,85,86]. The intermediate 11-oxo-THC can be transformed into 11-hydoxy-hydrocannabinol (11-hydroxy-HHC) and 11-nor-9-carboxy-HHC, compounds that can also be detected in urine [85]. All these metabolites will be glucuronoconjugated in phase 2 by the enzyme UDP-glucuronosyltransferase and can re-enter the enterohepatic circuit [87,88]. CBD is hydroxylated to 7-OH-CBD and then oxidized to 7-carboxy-CBD by CYP3A4 and CYP2C9 [89]. In the case of oral administration, there are studies that suggest that THC can be converted to CBD in the acidic gastric environment [90]. These transformations are schematically shown in Figure 6.

The half-life of THC can vary between 1.6 and 57 h, that of 11-hydroxy-THC between 12 and 36 h, and that of 11-nor-9-carboxy-THC between 1 and 6 days. After approximately 72 h, around 15% of THC is eliminated in the urine, and 50% in the feces. It is estimated that after 5 days, 80–90% of THC is eliminated from the body [80,91]. After a single dose, THC can be identified in the urine within a range of 1–6 h, 11-hydroxy-THC for 2–8 h and 11-nor-9-carboxy-THC from 2.9 days to 28 days, or even up to 67 days after repeated consumption [91,92,93]. CBD has a half-life of approximately 56–61 h [78] and a urinary detection period in the range of 30–52 h for both CBD and its metabolites, 7-OH-CBD and 7-carboxy-CBD [94].

### 4.3. Detection Methods for Abuse

The detection of phytocannabinoids is a challenge, both from the point of view of the complexity of the techniques used, and due to the particularities of each technique, with their advantages and disadvantages.

Electrochemical methods use reactions consisting in the electrochemical oxidation of THC in an alkaline solution, but in general, the determination of THC is quite difficult due to the hydrophobicity of the molecule [95,96]. Lateral-flow immunoassay using electrochemical transducers is a rapid method for the identification of THC which is able to detect it in an analysis time of 6 min with a detection limit of 1.3 ng/mL [97]. Another electrochemical method using an aptasensor for the detection of THC in saliva showed a one-minute identification [98] with a detection limit of 5 nM in serum, 5 nM in urine and 10 nM in 50% diluted saliva [99]. Despite the fact that these methods are very rapid and have a nanomolar detection limit, they cannot exclude false positive reactions during routine determinations, especially because they do not identify metabolites in the same run. The same drawback is represented by immunoassay-based methods where the interference of proton pump inhibitors, especially pantoprazole [100,101], efavirenz and nonsteroidal anti-inflammatory drugs [102] is well-known.

Phytocannabinoids can be identified by GC-MS techniques in both blood and urine as derivatized products [103,104,105,106,107]. Low detection limits range in urine from 10 ng/mL to 25 ng/mL for 11-nor-9-carboxy-THC, 1 ng/mL for THC and CBD, 5 ng/mL for 11-OH-THC [105,106] and in blood from 15 ng/mL to 20 ng/mL for THC and 40 ng/mL for 11-nor-9-carboxy-THC [106,107].

LC-MS/MS methods using different types of mass spectrometers as detectors, such as triple quadrupole, have been able to simultaneously identify a large number of natural cannabinoids and their metabolites, the main compounds being displayed in Figure 5. A total of 17 compounds including CBD, CBD metabolites, especially 7-OH-CBD, 7-carboxy-CBD, as well as glucuronidated variants could be highlighted in plasma [108]. Detection was performed both in positive ion mode detection and in negative ion mode detection. Low detection limits were between 0.5–2 ng/mL for CBD, THC, metabolites and glucuronidated compounds [109,110,111]. Urinary determinations showed a low detection limit of 1 ng/mL for non-carboxylated compounds and 5 ng/mL for carboxylated ones [112,113]. A multitude of hair cannabinoid analysis studies have been developed recently through which they can be identified in dynamics.

LC-MS [114,115,116,117,118] and GC-MS [119,120] hair identification methods have demonstrated their usefulness in presenting the “history” of a person’s consumption, but they have no applicability in the current context in sports competitions, because consumption is prohibited only during competition, recreational use being currently accepted by many countries.

## 5. Light/Beneficial Zone: Therapeutic Effects for Athletes

The other face of cannabinoids is the beneficial effect that is being progressively more highlighted with the increase in consumption due to legalization in many countries. In general, these effects are mediated through interactions with the cannabinoid receptors CB1 and CB2, but the process is much more complex because there are numerous additional interactions [121]. With new research, the use of substances contained in cannabis as dietary supplements is more and more being discussed, increasingly tending to become true nutrients [14,15,16]. Below, we will underline the positive effects of the numerous compounds found in the *Cannabis sativa* plant.

### 5.1. THC

The main psychoactive molecule, THC, may act as an inhibitor of moderate to severe chronic pain. In a study of 12 patients with neuropathic radicular pain in the lower limbs, Weizmen et al. [122] demonstrated that THC may induce pain relief through sympathovagal changes with a predominance of parasympathetic activity. THC has also been shown to have anti-inflammatory effects [123], such as in ulcerative colitis [124], and to have beneficial therapeutic effects in chronic pain [125,126]. The role of THC in reducing mortality in acute respiratory distress syndrome has been highlighted by attenuating the massive production and release of cytokines in the context of COVID-19 infection [127] and that mediated by Staphylococcal endotoxin B [128] by downregulating miR-185-3p expression in activated immune cells, by triggering apoptosis via the NFκB pathway [128]. Alvarez et al. injected 0.32 mg/kg THC, reducing the anti-inflammatory effect produced by HIV/SIV without producing xerostomia and psychotropic effects [129]. Despite this fact, due to its intense psychoactive effects, it has been less exploited for therapeutic uses, with the focus being on the other major cannabinoid, CBD.

### 5.2. CBD

CBD itself does not appear to influence sports performance, being considered safe from this point of view in several studies [130,131,132]. In general, the dose to obtain therapeutic effects varies in the range 160–800 mg for a single administration and 300–1200 mg for repeated administration [133].

In acute stress, CBD reduces anxious and depressive behavior [133], especially in anxiety associated with sports competitions [49]. These effects can also be explained by their binding as agonists to serotonin 5HT1A receptors, reducing depressive symptoms [134]. Several studies on patients diagnosed with generalized anxiety and panic attacks using CBD versus placebo showed a reduction in symptoms, including those related to sleep disturbances that accompanied these pathologies [135,136,137].

In cases of injuries caused by accidents or joint and muscle overtraining, an anti-inflammatory effect was evidenced by the decrease in plasma concentration of IL-6 (a proinflammatory cytokine) [138] without having an effect on TNF-α or IL-10 [139]. In weightlifters, back pain was alleviated by sublingual CBD administration, but this effect was not evident when compared to placebo in untrained individuals [48]. Another study highlighted the inhibitory effect of THC and CBD on cyclooxygenase 2 [140]. Several recent reviews [133,141] investigated the anti-inflammatory and analgesic effect of CBD and highlighted that there is no clinical evidence that CBD could be effective in acute pain but only in chronic pain, a situation frequently encountered in sports. The best results of using CBD in chronic joint inflammation are obtained in the case of topical application [133].

In one study, CBD was shown to accelerate wound healing when incorporated with 2-hydroxypropyl-B-cyclodextrin into a fibroin-based film. In this form, CBD enhanced cell migration and vascular endothelial growth factor expression in fibroblasts, an important phenomenon in tissue regeneration [142]. Another study found that CBD in alginate-based hydrogels improved wound healing by accelerating collagen fiber repair, promoting angiogenesis, and promoting granulation tissue [143]. CBD’s stimulatory effect on Connective Tissue Growth Factor also appears to play an important role in this process [144]. Other beneficial uses in terms of accelerating healing have been described in the treatment of superficial, isolated skin lesions (pustules, papules) or deep and severe lesions (epidermolysis bullosa, moderate to severe psoriasis) in local applications [145]. It seems that topical application of CBD preparations has a real effect on the healing of wounds, erosions, or other open injuries in athletes. As long as CBD is not a banned compound in sports competitions, its absorption from the application site is not an inconvenience in this regard.

### 5.3. Cannabigerol

Cannabigerol is a weak agonist of the CB1 receptor and a partial agonist of the CB2 receptor [146]. It may exhibit anti-inflammatory and antioxidant effects. The anti-inflammatory effect is explained by a reduction in the activity of myeloperoxidase and nitric oxide synthase and an increase in the activity of superoxide dismutase, while the antioxidant effect is produced by the reduction of reactive oxygen species via nitric oxide synthase, in a similar manner to that exerted by vitamin E [147,148]. It also reduces the expression of proinflammatory cytokines such as TNFα and IL-1β by reducing the activity of the nuclear factor NFκB [21].

In bone healing after fracture, cannabigerol showed an enhanced mineralization of soft callus and endochondral ossification through a pro-osteogenic effect in 14-week-old C57BL/6J mice [149]. A different study conducted on the same type of mice with osteoarthritis that received cannabigerol oil at a dose of 10 mg/kg/day reported reduced cartilage degeneration, improved synovitis, and a chondroprotective effect [150]. Another group of B6 and 129PF2 mice showed an improvement in locomotor activity and a reduction in anxiety when an extract containing cannabigerol was administered [151]. The anxiolytic effect was also highlighted by a double-blind study in which 37 adults received orally a tincture with 10 mg/mL cannabidiol, which resulted in a reduction in anxiety and stress compared to placebo [152]. Knowing that such incidents are quite common in athletes, the effects of cannabigerol on the osteoarticular system look promising and deserve to be investigated in further studies. Furthermore, cannabigerol has not demonstrated potential to improve athletic performance [153]. These aspects make it promising for use in sports injuries.

### 5.4. Monoterpenes

Monoterpenes, as a class of compounds, are found in a wide variety of plants. Of these, β-myrcene, α-pinene and limonene are best represented in *Cannabis sativa* [154].

β-myrcene, the main terpene, is mainly shown by its analgesic action produced through Transient Receptor Potential Ion Channel 1 (TRPV1) [155]. Other attributed effects are the sedative one which seems to depend on its content, being evident in the presence of large amounts [156]. β-myrcene also reduces joint inflammation by activating the cannabinoid receptors CB1 and CB2. However, its concomitant administration with CBD did not demonstrate a synergistic effect [157]. This is surprising because, in general, a synergistic effect, also called the “entourage effect”, has been described for most of the compounds present in *Cannabis sativa* [139].

Instead, pinene acts synergistically with the main cannabinoids, especially in the case of the anti-inflammatory and anxiolytic effect [158,159]. In this case, the “entourage effect” was demonstrated not by direct action on cannabinoid receptors, but indirectly, probably allosterically [160]. However, CB1 activation by THC is much reduced in the absence of pinene compared to its presence [161].

Of all the terpenes with the highest abundance in *Cannabis sativa*, limonene has not demonstrated anti-inflammatory properties [162]. By contrast, studies on the therapeutic effects of naturally occurring limonene, extracted from plants where it is found as a predominant component (such as citrus fruits), show anti-inflammatory [154] and neuroprotective [163] effects. However, in the case of *Cannabis sativa*, limonene may exert its pharmacological properties and contribute to global actions through an entourage effect [38,164].

### 5.5. Sesquiterpenoids

Sesquiterpenoids appear to act on CB2 receptors with anti-inflammatory action but without psychoactive effects. In the plant, this class of chemicals plays an antifungal and insect-repellent role [32]. Among the sesquiterpenoids, β-caryophyllene stands out in a study conducted on zebrafish as a real candidate for its use as a sedative [159].

### 5.6. Triterpenoids

The most abundant triterpenoid is friedelin followed by epifriedelanol. Both substances have demonstrated antioxidant properties using *Saccharomyces cerevisiae* as a model [165]. Friedelin demonstrated an anti-inflammatory action by inhibiting IL-1β production [166].

### 5.7. Sterols

Sterols, the most abundant of which are β-sitosterol, campesterol, and stigmasterol, are minor components of *Cannabis sativa*. β-sitosterol has presented anti-inflammatory properties [166]. In a recent study [167], it was shown that β-sitosterol from *Trema orientalis*, a member of the Cannabaceae family, increased the viability and growth of BF-2 cell lines (cells used as toxicological indicators). Thus, β-sitosterol could play a role as a growth factor used as a plant extract dietary supplement in the future.

### 5.8. Cannflavins

Cannflavins A, B and C are flavonoids specific to cannabis species that exhibit anti-inflammatory effects. There are several studies that attempt to elucidate the intimate mechanism of this anti-inflammatory effect. It has been demonstrated that cannflavin A and cannflavin B reduce the synthesis of PGE2 and leukotrienes by inhibiting the enzymes PGE synthase and 5-lipoxygenase [168], but also both cyclooxygenases 1 and 2 [169]. An important mechanism seems to be the mediation of the anti-inflammatory activity through toll-like receptor (TLR)4 which represents a molecular target for cannflavin A. This phenomenon has been observed at the level of macrophages, activation of TLR4 leading to the synthesis of chemokines CXCL10, IL-1β and TNFα. The action on TLR4 of cannflavin A induces a decrease in the production of CXCL10, IL-1β and TNFα [170]. The effect of cannflavins appears to be clinically important, as their anti-inflammatory action appears to be more intense than that produced by aspirin [171].

Another effect is the antioxidant one, with cannflavin A and cannflavin B playing a role in reducing lipid peroxidation, inactivating free radicals and inhibiting the synthesis of advanced glycated end products in human keratinocytes [172]. Thus, cannflavins appear as substances that could contribute to reducing inflammatory phenomena during training or prevent their occurrence in the event of joint burden.

## 6. Grey Zone: Discussions and Future Perspectives

The subject discussed in this review is a very contemporary one, which provokes interest in the context of changing legislation in many countries regarding the use of cannabis. In sports, things are much warmer in the context of balancing the positive effects produced by the compounds identified in *Cannabis sativa* and their negative effects, especially that of illegal sports performance enhancement.

There is practically no sport in which the use of compounds found in cannabis species produces the greatest benefits or the strongest dangerous effects; they all depend on the context. Chronic users are the most exposed to toxic effects due to THC with its addictive potential, but also through the negative effects exerted on the brain and heart. Of course, as it is known in toxicology that “the dose makes the difference”, the amount used is also important. On the other hand, the beneficial effects are also given by the severity of the injury, the psychological impact, the type of substance(s) used and also by the “entourage effect” between them.

In general, clinical data are rare in athletes, with no studies in this category of individuals, demonstrating, without conclusive evidence, anti-inflammatory, anxiolytic, antidepressant, or wound-healing effects, most of which are preclinical. In addition, no investigations have been conducted on cannabinoids during athletic competition; most conclusions are indirect [173]. Given the relaxation of the reporting requirements for THC and CBD, with the increase of the permitted THC concentration to 150 ng/mL and no CBD limitation [2], the following issues are raised into discussion:

If only CBD consumption is permitted, the use of CBD by a given athlete may generate traces of THC through the conversion of CBD to THC [90]. Furthermore, there is the possibility that 7-carboxy-CBD can be converted to 11-nor-9-carboxy-THC [174], generating false positive results. Studies on athletes on this subject are not yet complete and leave room for further research.Another discussion would be in the context of which beneficial effects occur in the case of natural extracts and to what extent they are attributed to the entourage effect [38,139]. Clinical results could be studied in the context of extraction and administration together and separately, and comparing the two aspects versus placebo in athletes. It is possible that the effect of CBD is enhanced by low doses of THC [38].Another aspect that needs to be considered is the perception of the sports physician about cannabinoids, both when and how they recommend them, knowing that THC accumulates in adipose tissue from where it is gradually released [78,80]. Here, too, it has been observed that young male physicians are more likely to accept the use of cannabinoids, not associating cannabis with the improvement of sports performance [175]. This aspect is of course in accordance with the legislation of different countries.Last but not least, when we talk about natural compounds, we are talking about extraction methods. The hypothetical discussion raised is related to the possibility of eliminating the THC compound from *Cannabis sativa* extract, which could make it possible to use the plant safely without interfering with legislative aspects.From another point of view, some athletes may find it convenient to cheat by relying on this legislative relaxation by consuming cannabis in order to achieve superior results in competitions. A recent study [176] aimed at finding out to what extent elite athletes (national, international competitions, Olympic or Paralympic games) dope, identified a percentage of 9.2% of athletes who consume prohibited substances during competitions. Of these, 4.2% are cannabinoid users. Of the cannabinoid users, 66.7% declared that they use two or more doping substances or methods. One of the limitations of this study lies in the honesty of the responses provided; the authors suspect that these data are actually underreported.

## 7. Conclusions

Doping in sports is a major current problem. With the legalization of recreational cannabis use, the issue of positive results in anti-doping controls complicates the context in which it occurs. However, ethical, moral, and legislative aspects remain, but not least those related to the adverse effects and the development of pathological aspects in chronic users.

On the other hand, there are arguments that support the beneficial effect of using cannabis products for anti-inflammatory, antioxidant, anxiolytic and accelerated wound healing in case of injuries or recovery and restoration after the competitive season. Among the cannabinoids, the most promising are CBD and cannabigerol for therapeutic use in athletes. Reports on cannflavins, which stand out among the minor components that may have an additional overall contribution, especially to the antioxidant and anti-inflammatory effects, should not be neglected.

With these paradigm shifts related to cannabinoids, there is an opportunity for studying these effects in active competitive athletes. These effects on athletes require a large study, carried out on large groups, by comparison with placebo, in order to be able to say with certainty whether these beneficial effects are real in athletes or are only indirectly deduced. Further research will be able to answer these questions in the future, while also establishing the boundary between abuse and therapy, a boundary that is currently extremely thin.

## Figures and Tables

**Figure 1 nutrients-17-00861-f001:**
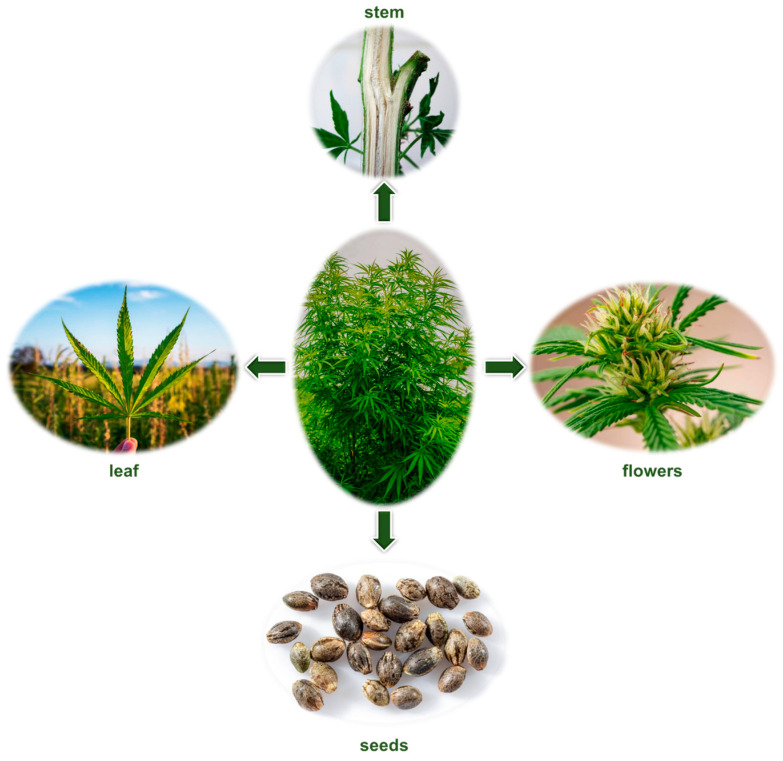
Diagram of a cannabis plant, illustrating its primary components: stem, leaves, flowers, and seeds (https://www.alamy.com/the-big-marijuana-tree-or-cannabis-cannabis-is-a-genus-of-flowering-plants-in-the-family-cannabaceae-the-number-of-species-within-the-genus-is-disputed-image380975038.html?imageid=8CC4A068-B453-4FFE-98BF-6452E78C29C5&p=1344718&pn=2&searchId=00ae21d2051aa197d038ea9207baf572&searchtype=0, accessed on 17 February 2025; https://www.alamy.com/hemp-or-marijuana-plants-growing-on-sunshine-at-farm-field-image372467727.html, accessed on 17 February 2025; https://www.alamy.com/close-up-of-a-cannabis-flower-in-full-bloom-flowering-marijuana-home-herb-cultivation-medical-marijuana-macro-image613742667.html?imageid=5A4B9837-AA8A-4B9D-9CF0-E4CBFAAFBCDC&p=811332&pn=1&searchId=87ce680caa70fe985ec5d14bb3d2412e&searchtype=0, accessed on 17 February 2025; https://en.wikipedia.org/wiki/Cannabis, accessed on 17 February 2025; https://www.alamy.com/cannabis-seeds-isolated-on-white-background-close-up-image444274200.html, accessed on 17 February 2025).

**Figure 2 nutrients-17-00861-f002:**
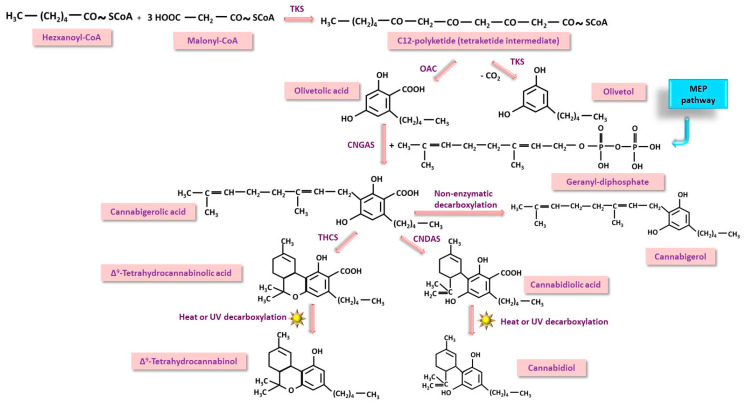
Polyketide pathway (PK) of cannabinoids biosynthesis in *Cannabis sativa*. One of the precursors, geranyl-diphosphate is a compound formed in both MEP and MV pathways. TKS—tetraketide synthase; OAC—olivetolic acid cyclase; CNGAS—cannabigenolic acid synthase; THCS—tetrahydrocannabinoid synthase; CNDAS—cannabidiolic acid synthase.

**Figure 3 nutrients-17-00861-f003:**
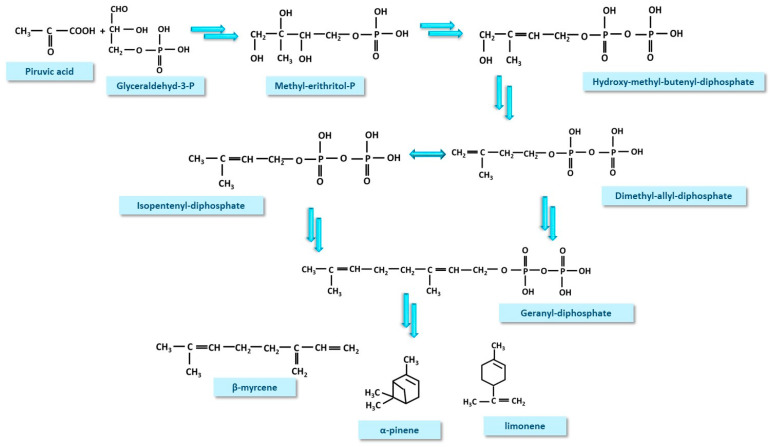
Methyl-erythritol phosphate (MEP) pathway displays, in the first phase, a common route to obtain geranyl-diphosphate. Geranyl-diphosphate serves further as precursor for all routes, except the phenyl-propanoid (PP) pathway. The final products of MEP are monoterpenoids.

**Figure 4 nutrients-17-00861-f004:**
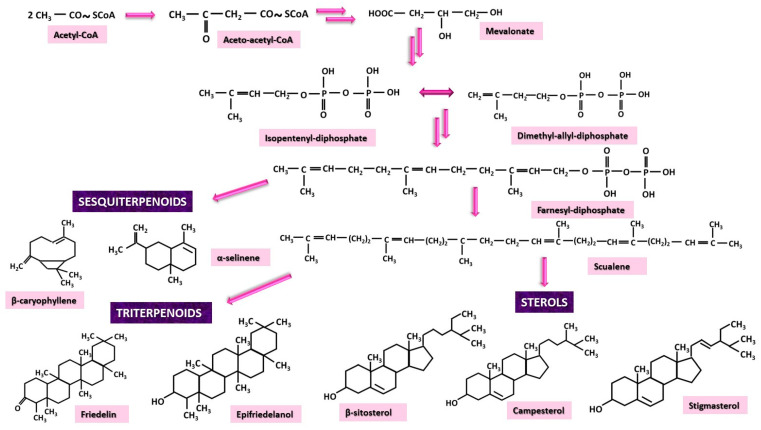
The mevalonate (MV) pathway has three categories of substances as final products: sesquiterpenoids, triterpenoids and sterols.

**Figure 5 nutrients-17-00861-f005:**
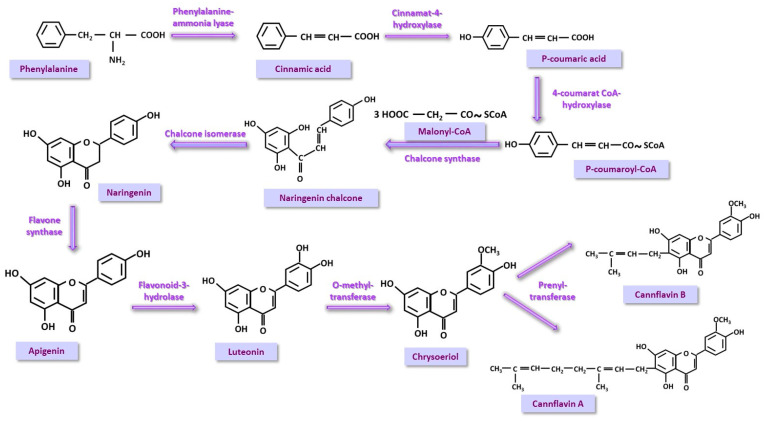
The phenyl-propanoid (PP) pathway is a natural route of flavonoid synthesis. The formation of cannflavins, specific to *Cannabis sativa*, is presented.

**Figure 6 nutrients-17-00861-f006:**
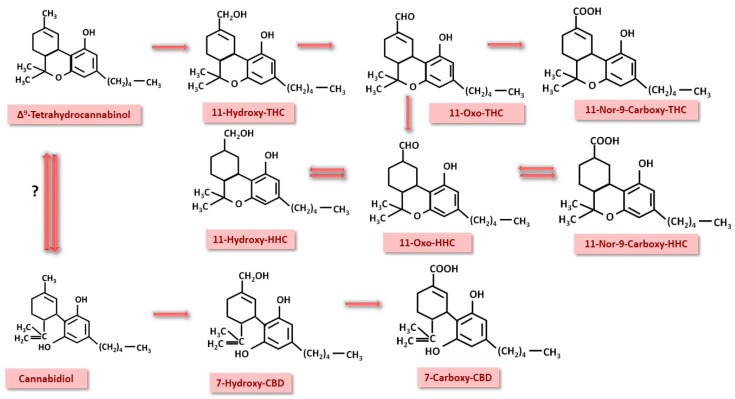
Phase 1 hepatic metabolism of THC and CBD. The resulting products are identified in urine in a routine toxicologic screening. THC—Δ^9^-tetrahydrocannabinol; HHC—hydrocannabinol; CBD—cannabidiol.

**Table 1 nutrients-17-00861-t001:** Brief overview of the legislative situation for the use of cannabinoids in several countries.

Country	Legalization, Comments	Study
United States	38 states for medicinal use 23 states for recreational use	Tavabi N et al., 2023 [67]
Nederlands	Recreational since 1967	Knottnerus JA 2023 [68]
Switzerland	Medical use since 2022	Palmiere C et al., 2024 [69]
Germany	Medical use since 2017 Recreational since 2024	Klosterkotter J et al., 2024 [70] Thomasius R et al., 2024 [71]
Canada	Medical use since 2001 Recreational since 2018	Armstrong MJ, 2020 [72]
Sweden	Medical: CBD only Recreational: forbidden	Feltmann K et al., 2024 [73] Helander, A et al., 2024 [74]
Serbia Bulgaria Nepal Malaysia Iran Kenya Ethiopia	Medical and Recreational: forbidden	Tihauan BM et al., 2025 [75] Ransing R et al., 2023 [76]
France Poland UK Romania Hungary Norway Finland	Medical use with prescription: legal Recreational use: illegal	Tihauan BM et al., 2025 [75] Ransing R et al., 2023 [76]
Thailand	Medical use since 2019 Recreational since 2021	Kalayasiri, R et al., 2023 [77]
Spain	Medical use since 2010 Recreational: allowed only for personal use	Ransing R et al., 2023 [76]
France	Medical use since 2020 Recreational: illegal	Ransing R et al., 2023 [76]

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
