# Peer review of "Cannabis: Zone Aspects of Raw Plant Components in Sport—A Narrative Review"

_nutrients, 2025, doi:10.3390/nu17050861_

Round 1

Reviewer 1 Report

Comments and Suggestions for Authors

I congratulate the authors for submitting a manuscript on such an emergent and interesting topic. I invite them to make some revisions before it can be published in Nutrients.

The type of review (narrative) has to be mentioned in the title, in the abstract, and in the whole manuscript.

The abstract is not proper. It should be structured in Background/Objectives; Methods (the applied methodologies such as searched databases, inclusion/exclusion criteria, and keywords used in the search need to be provided); Results; Conclusions, and future perspectives.

The section 1 (Introduction) is very poor. Much more is expected from an introductory section of a review manuscript submitted to an international Q1 journal like Nutrients. More citations and more detailed data about the topics to be addressed must be given; The rationale and justification to carry out the study have to be clarified, as well as its novelty. Can you please also include an illustration of the Cannabis plant in this section?

A Methods section is missing. See the example of section 2 of this published paper: https://www.mdpi.com/2304-8158/10/6/1175

Increase the size and improve the quality of the figures.

Can you please provide a table summarizing the point of the situation regarding the use and applications of cannabinoids in different regions/countries worldwide?

I would like to see a separate section discussing the legislation context worldwide, what is being done regarding that, and what should be done.

In which sports the use of the Cannabis plant and in which form(s) would bring more benefits and dangers? 

Author Response

I congratulate the authors for submitting a manuscript on such an emergent and interesting topic. I invite them to make some revisions before it can be published in Nutrients.

The type of review (narrative) has to be mentioned in the title, in the abstract, and in the whole manuscript.

The abstract is not proper. It should be structured in Background/Objectives; Methods (the applied methodologies such as searched databases, inclusion/exclusion criteria, and keywords used in the search need to be provided); Results; Conclusions, and future perspectives.

The section 1 (Introduction) is very poor. Much more is expected from an introductory section of a review manuscript submitted to an international Q1 journal like Nutrients. More citations and more detailed data about the topics to be addressed must be given; The rationale and justification to carry out the study have to be clarified, as well as its novelty. Can you please also include an illustration of the Cannabis plant in this section?

A Methods section is missing. See the example of section 2 of this published paper: https://www.mdpi.com/2304-8158/10/6/1175

Increase the size and improve the quality of the figures.

Can you please provide a table summarizing the point of the situation regarding the use and applications of cannabinoids in different regions/countries worldwide?

I would like to see a separate section discussing the legislation context worldwide, what is being done regarding that, and what should be done.

In which sports the use of the Cannabis plant and in which form(s) would bring more benefits and dangers? 

Response:

  1. The type of review was specified in the introduction and in methods sections, as sugested by the referee

„In this narrative review......”

  1. The abstract was improved as referee sugested:

Objectives/Background: The Cannabis genus includes a series of plants that contain a mixture of cannabinoids as major components and a heterogeneous group of other substances as minor components that have been explored over time. In this review, we aimed to highlight the main aspects of the polarized discussion between abuse and toxicity versus the benefits of the compounds found in the Cannabis sativa plant. Methods: We searched electronic databases such as PubMed, Google Scholar and Web of Science as well as World Anti-doping Agency (WADA) documents for scientific publications that can elucidate the heated discussion related to the negative aspects of addiction, organ damage and improved sports performance in the case of chronic users and the medical benefits, particularly in athletes, of some compounds that may be promising as micronutrients. Results: Scientific arguments are brought forward on the harmful effects of cannabinoids, especially Δ9-tetrahydrocannabinol, ethical and legislative aspects in the case of their use as doping substances in sports, but also their beneficial effects. In the case of cannabis consumption, not only the Δ9-tetrahydrocannabinol compound is of interest but also the other substances with which it is mixed. On the other hand, there are numerous studies that attest the beneficial effects, especially the anti-inflammatory, antioxidant, anxiolytic ones, which could bring a therapeutic advantage to athletes in case of injuries. Furthermore, we presented the generation of the main cannabis compounds and their transformation in the body into final products along with identification methods for routine anti-doping tests. The other aspect of the benefits, leaves room for the use of compounds extracted from Cannabis sativa to be used as micronutrients but also for their study as potential pharmacological agents. Conclusions and future perspectives:  From the perspective of both athletes and those who have identified illegal use in sport, there are many variable aspects that lead to many interpretations, presented and discussed in this review. Even though there are many recent studies on cannabis species, there is very little research on the beneficial effects in active athletes, and those on large groups of athletes compared to placebo are lacking. These studies are necessary to complete the current vision of this topic and to clarify the hypotheses launched as discussions in this review.”

  1. Introduction was improved and enlarged as well as references were inserted accordingly as referee suggested. An image with Cannabis sativa plant was added as Figure 1 according the referee
  2. A „Methods” section was introduced:

„In this narrative review, we used the databases PubMed, Google Scholar and Web of Science as well as WADA documents, and selected the most relevant papers for the purpose of this review. We used as keywords "cannabinoids", "sport", "illegal use", "tetrahydrocannabinol", "cannabidiol", "cannabigerol", "myrcene", "limonene", "pynene", "sesquiterpenoids", "friedelin", "sitosterol", "cannflavins", "pharmacokinetics" and combinations between them. The toxic, moral and illegal effects (dark zone) of cannabinoid consumption by athletes were discussed in parallel with the beneficial effects (light zone) and the possibility of using these compounds as micronutrients. Between the two opposite zones, sensitive and questionable aspects of this consumption (grey zone) were also touched upon.”

  1. In response to the reviewer's request to increase the quality of the images, we have attached previously these images separately so that they can be taken over by the publisher and arranged as best as possible.
  2. We introduced a table, named Table 1 regarding cannabis situation in some countries, as required by the referee
  3. We discussed the worldwide legislation in a new paragraph „4.1. A brief discussion of current worldwide legislative implications” as required by the referee
  4. We introduced a comment in the last section, in response to referee suggestion:

„There is practically no sport in which the use of compounds found in cannabis species produces the greatest benefits or the strongest dangerous effects, they all depend on the context. Chronic users are the most exposed to toxic effects due to THC with its addictive potential but also through the negative effects exerted on the brain and heart. Of course, as is known in toxicology that "the dose makes the difference", the amount used is also important. On the other hand, the beneficial effects are also given by the severity of the injury, the psychological impact, the type of substance(s) used and also by the "entourage effect" between them.”

Reviewer 2 Report

Comments and Suggestions for Authors

Thank you for summarizing the actual cannabis discussion from the scientific base to the clinical and ethical aspects. I would suggest to focus the paper on cannabis as nutritional supplement. So your article woul get into the scope and topic of Nutrients.

Author Response

Thank you for summarizing the actual cannabis discussion from the scientific base to the clinical and ethical aspects. I would suggest to focus the paper on cannabis as nutritional supplement. So your article would get into the scope and topic of Nutrients.

Response:

As recommended, in the abstract, introduction section as well as in the section 5, we have nuanced the role of cannabis compounds as nutrients.

Reviewer 3 Report

Comments and Suggestions for Authors

paper title - Cannabis: zone aspects of raw plant components in sport

here are some questions - comments, and recommendations

in the abstract - for review studies, typically abstract would contain the scope or range of the review, for instance... how many studies from ....what year to what year were reviewed....

introduction can be expanded to include more information as provided in the abstract - which is much clear

perhaps cite some policies or laws...

section 2 seems fine

was not able to find the method section - this section should note how the review is done

.... typically author/s would make use of PRISMA method

the rest of the findings seems fine, however, these findings should be supported by the method section

Author Response

in the abstract - for review studies, typically abstract would contain the scope or range of the review, for instance... how many studies from ....what year to what year were reviewed....
introduction can be expanded to include more information as provided in the abstract - which is much clear
perhaps cite some policies or laws...
section 2 seems fine
was not able to find the method section - this section should note how the review is done
.... typically author/s would make use of PRISMA method
the rest of the findings seems fine, however, these findings should be supported by the method section

Response:

  1. The abstract was improved as referee suggested:

Objectives/Background: The Cannabis genus includes a series of plants that contain a mixture of cannabinoids as major components and a heterogeneous group of other substances as minor components that have been explored over time. In this review, we aimed to highlight the main aspects of the polarized discussion between abuse and toxicity versus the benefits of the compounds found in the Cannabis sativa plant. Methods: We searched electronic databases such as PubMed, Google Scholar and Web of Science as well as World Anti-doping Agency (WADA) documents for scientific publications that can elucidate the heated discussion related to the negative aspects of addiction, organ damage and improved sports performance in the case of chronic users and the medical benefits, particularly in athletes, of some compounds that may be promising as micronutrients. Results: Scientific arguments are brought forward on the harmful effects of cannabinoids, especially Δ9-tetrahydrocannabinol, ethical and legislative aspects in the case of their use as doping substances in sports, but also their beneficial effects. In the case of cannabis consumption, not only the Δ9-tetrahydrocannabinol compound is of interest but also the other substances with which it is mixed. On the other hand, there are numerous studies that attest the beneficial effects, especially the anti-inflammatory, antioxidant, anxiolytic ones, which could bring a therapeutic advantage to athletes in case of injuries. Furthermore, we presented the generation of the main cannabis compounds and their transformation in the body into final products along with identification methods for routine anti-doping tests. The other aspect of the benefits, leaves room for the use of compounds extracted from Cannabis sativa to be used as micronutrients but also for their study as potential pharmacological agents. Conclusions and future perspectives:  From the perspective of both athletes and those who have identified illegal use in sport, there are many variable aspects that lead to many interpretations, presented and discussed in this review. Even though there are many recent studies on cannabis species, there is very little research on the beneficial effects in active athletes, and those on large groups of athletes compared to placebo are lacking. These studies are necessary to complete the current vision of this topic and to clarify the hypotheses launched as discussions in this review.”

  1. Introduction was improved and enlarged as well as references were inserted accordingly as referee suggested.
  2. We discussed the worldwide legislation in a new paragraph „4.1. A brief discussion of current worldwide legislative implications” as required by the referee
  3. A „Methods” section was introduced:

„In this narrative review, we used the databases PubMed, Google Scholar and Web of Science as well as WADA documents, and selected the most relevant papers for the purpose of this review. We used as keywords "cannabinoids", "sport", "illegal use", "tetrahydrocannabinol", "cannabidiol", "cannabigerol", "myrcene", "limonene", "pynene", "sesquiterpenoids", "friedelin", "sitosterol", "cannflavins", "pharmacokinetics" and combinations between them. The toxic, moral and illegal effects (dark zone) of cannabinoid consumption by athletes were discussed in parallel with the beneficial effects (light zone) and the possibility of using these compounds as micronutrients. Between the two opposite zones, sensitive and questionable aspects of this consumption (grey zone) were also touched upon.”

Round 2

Reviewer 1 Report

Comments and Suggestions for Authors

The type of review (narrative) has to be mentioned in the title, in the abstract, and in the whole manuscript. This comment was not addressed by the authors. The type of review (narrative) has to be mentioned in the title and the abstract.

The abstract is too extensive. The word limit is 250.

Author Response

The type of review (narrative) has to be mentioned in the title, in the abstract, and in the whole manuscript. This comment was not addressed by the authors. The type of review (narrative) has to be mentioned in the title and the abstract.
The abstract is too extensive. The word limit is 250.

Response:

  1. The type of reviw was specified in the title and abstract, as sugested by the referee

Cannabis: zone aspects of raw plant components in sport - a narrative review”

„In this narrative review......”

  1. The abstract was modified and does not exceed the 250 word limit:

Objectives/Background: The Cannabis genus contain a mixture of cannabinoids and other minor components studied so far. In this narrative review, we highlight the main aspects of the polarized discussion between abuse and toxicity versus the benefits of the compounds found in the Cannabis sativa plant. Methods: We investigated databases such as PubMed, Google Scholar, Web of Science and World Anti-doping Agency (WADA) documents for scientific publications that can elucidate the heated discussion related to the negative aspects of addiction, organ damage and improved sports performance and the medical benefits, particularly in athletes, of some compounds that are promising as nutrients. Results: Scientific arguments bring forward the harmful effects of cannabinoids, ethical and legislative aspects of their using as doping substances in sports. Hence, we presented the synthesis and metabolism of the main cannabis compounds along with identification methods for routine anti-doping tests. Other numerous studies attest the beneficial effects, which could bring a therapeutic advantage to athletes in case of injuries. Their benefits, recommends Cannabis sativa compounds as nutrients, as well as potential pharmacological agents. Conclusions and future perspectives:  From the perspective of both athletes and illegal use investigators in sport, there are many interpretations, presented and discussed in this review.  Despite of many recent studies on cannabis species, there is very little research on the beneficial effects in active athletes, especially on large groups compared to placebo. These studies may complete the current vision of this topic and clarify the hypotheses launched as discussions in this review”

Reviewer 2 Report

Comments and Suggestions for Authors

Thank you for improving the manuscript and focusing the paper.

Reviewer 3 Report

Comments and Suggestions for Authors

After going over the revisions made

the paper is now suitable for acceptance

Author Response

After going over the revisions made the paper is now suitable for acceptance

Response:

Dear reviewer, we would like to thank you for you significant contribution to improve our manuscript!